# LOST IN TRANSFORMATION: CURRENT ROADBLOCKS FOR TRANSFORMERS IN 3D MEDICAL IMAGE SEGMENTATION

## ABSTRACT

In the medical image segmentation domain, sparsely-annotated, limited datasets are common, posing a natural hurdle for Transformer-based segmentation networks. In this work, we systematically dissect 9 such popular Transformer networks on two representative organ and pathology segmentation datasets and explore whether Transformers are still beneficial under these challenging conditions. **1)** We demonstrate that these Transformer-based segmentation networks frequently incorporate substantial convolutional backbones, which predominantly contribute to their performance, while Transformers themselves play a peripheral role. **2)** Extending beyond accuracy, we analyze error and representational similarity to uncover architectures with underutilized Transformers, demonstrated by indiscernible change on both metrics *without* the Transformer. **3)** We quantify the massive dataset size 'chasm' between medical and natural images, examine the impact of data reduction on performance, showing that Transformers bridge the performance gap to CNNs as the dataset size increases. **4)** Additionally, we probe the importance of long-range interactions, showing that even limited receptive fields offer *high* performance in segmenting medical images, questioning the need for long-range interactions inherent to Transformers. In doing so, we identify significant challenges faced by major architectures employing Transformers for medical image segmentation, which may contribute to potential inefficiencies downstream in the domain.

## 1 INTRODUCTION

In recent years, attention mechanisms have taken center stage across various research domains (Chang et al. (2023); Ahmed et al. (2023); Aleissaee et al. (2023); Khan et al. (2022); Zhang et al. (2023). Originally proposed for natural language processing by (Vaswani et al. (2017)), the attention mechanism has played a pivotal role in advancing these fields. Within the realm of medical image analysis, semantic segmentation is a critical challenge (see §A) which has also seen a significant influx of Transformer-based semantic segmentation architectures that heavily rely on attention mechanisms (Xiao et al. (2023); Shamshad et al. (2023)). These architectures combine elements from Vision or Swin Transformers (Dosovitskiy et al. (2020); Liu et al. (2021b)) and Convolutional Neural Networks (ConvNets), occasionally exhibiting similarities to the structure of a UNet (Ronneberger et al. (2015)). However, there exist two fundamental distinctions between training networks for natural language (or natural images) and medical images: 1) Medical image segmentation networks typically do not undergo pretraining on extensive datasets (Radford et al. (2021); Dosovitskiy et al. (2020)). 2) The target datasets for medical images usually contain significantly fewer samples and sparse annotations (Litjens et al. (2017)). While the influence of these two massive caveats are implicitly recognized, the domain of medical image segmentation typically overlooks a pivotal question:

> Do the advantages of Transformer-based networks translate to *severely limited*, *sparsely-annotated* medical image segmentation datasets when *trained from scratch*?

In this work, we dissect nine influential Transformer-based networks employed for medical image segmentation. Our fundamental objective is to better understand how the inclusion of Transformers in these models affects their performance. In the process, we uncover a range of potential issues inherent to these architectures within the medical imaging domain.

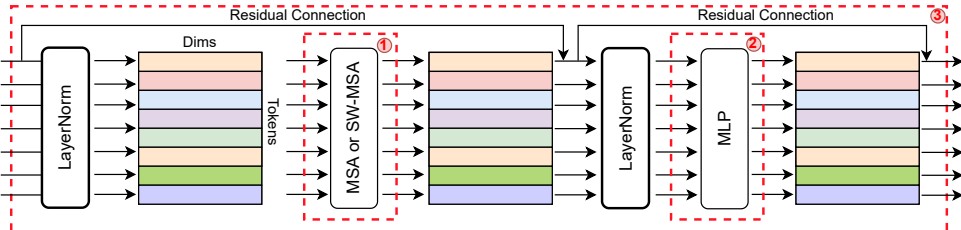

Figure 1: **Segmentation performance pre-and-post Identity replacement of a Transformer module quantifies their importance.** There are **3 points** of replacement on Vision or Swin Transformer blocks: **1)** Attention replacement, **2)** MLP Replacement, **3)** Whole Transformer Replacement.

1. We demonstrate that an influential (and often large) Convolutional (ConvNet) backbone exists in almost all our Transformer-based medical image segmentation architectures. When we replace these Transformers (§2), the backbone largely maintains segmentation performance, indicating a marginal role of the Transformer (§3).

2. We explore the similarity of errors and representations learned by the ConvNet backbone and find architectures with similar errors as well as representations, regardless of the presence of Transformers, indicating *neglible utilization of the Transformer*. In others, we interestingly observe differences in representation learning but *without* differences in accuracy (§4).

3. We quantify the domain 'chasm' between medical and natural images, showing that Transformer-based networks are *less efficient at utilizing limited training data* common to medical image datasets – pure ConvNets being demonstrably better at doing so (§5.1,§5.2).

4. While Transformers are recognized for their capacity to capture long-range dependencies, we offer a counterexample which emphasizes that such dependencies may not be as critical for medical image segmentation as previously assumed. We demonstrate the diminishing advantages of employing larger receptive fields for achieving high performance, questioning the necessity of Transformer-based architecture designs. (§5.3).

In doing so, we are the first publication to put a spotlight on current roadblocks inhibiting Transformer-based networks from outperforming CNN-based medical image segmentation architectures.

## 2 EXPERIMENTAL DESIGN

**Standard Networks** We dissect a number of massively-influential Transformer architectures for medical image segmentation, regularly used as blueprints for designing newer architectures or state-of-the-art baselines. In this work, we focus on 9 such networks with *5500+ citations* collectively in the last 3 years (see Table 5): (i) UNETR (Hatamizadeh et al. (2022)) (ii) SwinUNETR (Hatamizadeh et al. (2021)) (iii) CoTr (Xie et al. (2021)) (iv) TransFuse (Zhang et al. (2021)) (v) nnFormer (Zhou et al. (2021; 2023)) (vi) SwinUNet (Cao et al. (2022)) (vii) UTNet (Gao et al. (2021)) (viii) TransBTS (Wang et al. (2021)) (ix) TransUNet (Chen et al. (2021)).

**Network Modification Scheme** Owing to large variations in hybridized Transformer and CNN architectures, we seek to investigate which architecture offers the best integration of attention. To this end, we design *network modification experiments* to gauge the influence of Transformer components on overall network performance in established medical image segmentation architectures. We evaluate replacing the 1) Attention, 2) Transformer block MLP and 3) Whole Transformer. The first two of these represent substructures of the overall Transformer while the last one removes the entire Transformer block, as illustrated in Fig. 1. We use *Identity* blocks as drop-in replacements of the respective Transformer block, which simply pass the representations to the next block, leaving the remaining architecture structure untouched. We perform this modification for all Transformer blocks throughout the entire architecture for all 9 networks.

**Training and Datasets** We follow the most popular technique of training deep neural networks for medical image segmentation task – training from scratch on a task-specific dataset. We leverage

| | TransBTS | TransFuse | TransUNet | UNETR | UTNet | CoTR | nnFormer | SwinUNet | SwinUNETR |
|---|---|---|---|---|---|---|---|---|---|
| Input Dims | 3D | 2D | 2D | 3D | 2D | 3D | 3D | 2D | 3D |
| **Used in Encoder** | | | | | | | | | |
| Convolutions | + | + | + | - | + | + | Down | - | - |
| Attention | - | SA | - | SA | SA | DSA | Swin | Swin | Swin |
| **Used in Decoder** | | | | | | | | | |
| Convolutions | + | + | + | + | + | + | Up | - | + |
| Attention | - | - | - | - | SA | - | Swin | Swin | - |
| **Used in Bottleneck** | | | | | | | | | |
| Convolutions | - | - | - | + | + | + | Up & Down | - | + |
| Attention | SA | - | SA | - | SA | - | Swin | Swin | - |
| Direct Conv ⟶ Out | + | + | + | Highest-res | - | - | - | - | Highest-res |
| Blocks Intermingled | - | - | - | - | + | - | + | - | - |
| **Parameters** | | | | | | | | | |
| Total [e+06] | 31.6 | 26.4 | 105.9 | 92.8 | 10.0 | 41.9 | 38.1 | 41.4 | 62.2 |
| WholeBlock [%] | 66.5% | 53.8% | 80.3% | 91.6% | 25.6% | 22.2% | 61.5% | 91.2% | 7.9% |
| Attn. [%] | 13.3% | 17.9% | 26.8% | 30.5% | 19.5% | 5.3% | 20.4% | 30.5% | 2.8% |
| MLP [%] | 53.2% | 35.8% | 53.5% | 61.1% | 4.3% | 16.9% | 41.1% | 60.7% | 5.0% |
| Convs [% of UNet[1]] | 65% | 39% | 67% | 48% | 24% | 200% | 90% | 12% | 352% |

Table 1: **Transformer-based networks often have significant Convolutional backbones.** Upon closer inspection, 8 out of 9 architectures make extensive use of convolutions (24-352% of a UNet[1]) – which are fundamental to their segmentation performance (Table 2). [**(D)SA:** (Deformable) Self-Attention, **Swin:** Shifted-Window Attention, **Up, Down:** Convolutions only used in up or downsampling, **Highest-res:** Direct non-Transformer input to output path only exists at the highest resolution.]

the established nnUNet framework (Isensee et al. (2021)) for training and evaluation. For each experiment, we train 3 folds of the same model to account for training noise and provide average performance on a hold-out testset. Further details are provided in §B. In terms of datasets, we distinguish between pure organ segmentation and pathology focused segmentation. In this work, we choose one large dataset of each category, to represent the majority of use-cases represented in the domain. We chose the **AMOS Abdominal Multi-Organ CT** (Ji et al. (2022)) and **Kidney Tumor Challenge 2019 (KiTS19)** (Heller et al. (2021)) datasets for the purposes of our evaluation. In the original works of the to-be-dissected architectures, organ segmentation is used more prominently than pathology segmentation (Fig. 3). However, pathologies are generally heterogeneous structures and their accurate fully automatic segmentation is considered a more challenging task than organ segmentation (Ghaffari et al. (2019); Heller et al. (2021); Zhong et al. (2022); Bilic et al. (2023)).

# 3 CONVNETS IN DISGUISE: THE INFLUENTIAL CONVOLUTIONAL BACKBONE

All Transformer-based semantic segmentation architectures employ attention mechanisms in varying ways. In Table 1, we provide an overview of the network characteristics of our Transformer-based architectures, revealing a consistent presence of a ConvNet backbone in nearly all of them. Notably, our examination shows that among the nine architectures, five incorporate convolutions in the encoder, while all but one incorporate some form of convolution in the decoder. Regarding the distribution of learnable parameters, we find that seven of these architectures allocate over 40% of their parameters within the Transformer blocks. Notably, both SwinUNet and UNETR allocate more than 90% of their parameters to these blocks. While this may appear to be a substantial portion of the parameters, it is informative to contextualize the absolute number of trainable parameters that exist outside the Transformer blocks, primarily within the convolution layers. A meaningful comparison can be drawn by contrasting this with the total parameter count in a standard 2D UNet and 3D-UNet[1]. Remarkably, all our Transformer-based 3D architectures contain at least 48% and at most 352% as many convolution parameters as a standard 3D-UNet. Similarly, in the case of 2D networks, they utilize between 12% and 67% of the convolution parameters found in a standard 2D UNet. This fact becomes particularly noteworthy when considering the individual segmentation performance of this backbone, independently of the Transformer components.

## 3.1 CONVNET BACKBONES DRIVE SEGMENTATION PERFORMANCE

Having established the presence of a significant ConvNet backbone in each network in §3, we measure Transformer effectiveness *by omission* and ask the following question: *How effective is this*

---

[1]We use *standard UNet* as proxy for original 2D/3D UNet (Ronneberger et al. (2015); Çiçek et al. (2016))

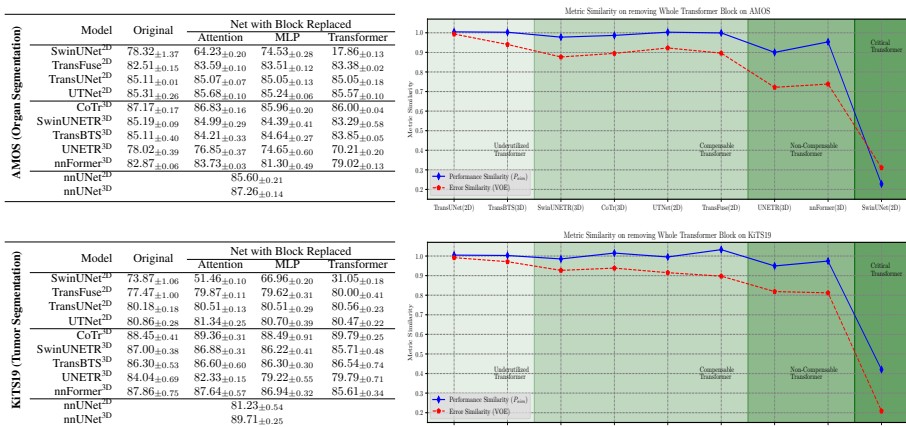

| | Model | Original | Net with Block Replaced | | |
| --- | --- | --- | --- | --- | --- |
| | | | Attention | MLP | Transformer |
| AMOS (Organ Segmentation) | SwinUNet[2D] | $78.32_{\pm1.37}$ | $64.23_{\pm0.20}$ | $74.53_{\pm0.28}$ | $17.86_{\pm0.13}$ |
| | TransFuse[2D] | $82.51_{\pm0.15}$ | $83.59_{\pm0.10}$ | $83.51_{\pm0.12}$ | $83.38_{\pm0.02}$ |
| | TransUNet[2D] | $85.11_{\pm0.01}$ | $85.07_{\pm0.07}$ | $85.05_{\pm0.13}$ | $85.05_{\pm0.18}$ |
| | UTNet[2D] | $85.31_{\pm0.26}$ | $85.68_{\pm0.10}$ | $85.24_{\pm0.06}$ | $85.57_{\pm0.10}$ |
| | CoTr[3D] | $87.17_{\pm0.17}$ | $86.83_{\pm0.16}$ | $85.96_{\pm0.20}$ | $86.00_{\pm0.04}$ |
| | SwinUNETR[3D] | $85.19_{\pm0.09}$ | $84.99_{\pm0.29}$ | $84.39_{\pm0.41}$ | $83.29_{\pm0.58}$ |
| | TransBTS[3D] | $85.11_{\pm0.40}$ | $84.21_{\pm0.33}$ | $84.64_{\pm0.27}$ | $83.85_{\pm0.05}$ |
| | UNETR[3D] | $78.02_{\pm0.39}$ | $76.85_{\pm0.37}$ | $74.65_{\pm0.60}$ | $70.21_{\pm0.20}$ |
| | nnFormer[3D] | $82.87_{\pm0.06}$ | $83.73_{\pm0.03}$ | $81.30_{\pm0.49}$ | $79.02_{\pm0.13}$ |
| | nnUNet[2D] | $85.60_{\pm0.21}$ | | | |
| | nnUNet[3D] | $87.26_{\pm0.14}$ | | | |
| KiTS19 (Tumor Segmentation) | SwinUNet[2D] | $73.87_{\pm1.06}$ | $51.46_{\pm0.10}$ | $66.96_{\pm0.20}$ | $31.05_{\pm0.18}$ |
| | TransFuse[2D] | $77.47_{\pm1.00}$ | $79.87_{\pm0.11}$ | $79.62_{\pm0.31}$ | $80.00_{\pm0.41}$ |
| | TransUNet[2D] | $80.18_{\pm0.18}$ | $80.51_{\pm0.13}$ | $80.51_{\pm0.29}$ | $80.56_{\pm0.23}$ |
| | UTNet[2D] | $80.86_{\pm0.28}$ | $81.34_{\pm0.25}$ | $80.70_{\pm0.39}$ | $80.47_{\pm0.22}$ |
| | CoTr[3D] | $88.45_{\pm0.41}$ | $89.36_{\pm0.31}$ | $88.49_{\pm0.91}$ | $89.79_{\pm0.25}$ |
| | SwinUNETR[3D] | $87.00_{\pm0.38}$ | $86.88_{\pm0.31}$ | $86.22_{\pm0.41}$ | $85.71_{\pm0.48}$ |
| | TransBTS[3D] | $86.30_{\pm0.53}$ | $86.60_{\pm0.60}$ | $86.30_{\pm0.30}$ | $86.54_{\pm0.74}$ |
| | UNETR[3D] | $84.04_{\pm0.69}$ | $82.33_{\pm0.15}$ | $79.22_{\pm0.55}$ | $79.79_{\pm0.71}$ |
| | nnFormer[3D] | $87.86_{\pm0.75}$ | $87.64_{\pm0.57}$ | $86.94_{\pm0.32}$ | $85.61_{\pm0.34}$ |
| | nnUNet[2D] | $81.23_{\pm0.54}$ | | | |
| | nnUNet[3D] | $89.71_{\pm0.25}$ | | | |

Table 2: **ConvNet backbones account for the majority of segmentation performance, while Transformers have a peripheral role.** (Left) 8 out of 9 networks show >90% performance similarity ($\mathbf{P_{sim}}$), while some retain upto 99% on both KiTS19 and AMOS without the entire Transformer. (Right) We additionally use error similarity (**VEO**) to categorize the different Transformers (§4.1).

*ConvNet backbone **without** the Transformer* for learning useful representations for 3D medical image segmentation? Through our network modification scheme (Fig. 1), we replace the whole Transformer blocks (as well as Attention and MLP separately) with an identity mapping, allowing us to measure the contribution of the remaining blocks. Across our network modification experiments (Table 2), we report relative performance similarity ($P_{sim}$) based Dice Similarity Coefficient (DSC)(Zijdenbos et al. (1994)) with a Transformer present and absent and notice a persistently similar yet *significant* observation on the lack of influence of Transformer blocks on segmentation performance, as well as one interesting counter-example to this observation.

1. **ConvNet backbones drive performance:** In 8 out of 9 architectures on both datasets, ConvNet backbones significantly (and sometimes completely) compensate for the *absolute lack of a Transformer component*. In fact, for the AMOS dataset, 6 of those do not have a 2% drop in performance while 4 do not have a 1% reduction. For the more challenging KiTS19 dataset, 5 out of 8 do not show a 3% drop in segmentation performance. This highlights the undue influence the hidden *ConvNet backbone* possesses in standard medical image segmentation, independent of the presence of a Transformer block. Additionally, this can be a significant insight for designing derived architectures – if the Transformer model does not significantly enhance performance, minor tweaks may also not yield substantial improvements in performance, while preserving the core Transformer-based design.

2. **Transformers are not completely *incapable* of learning:** The only network that catastrophically degrades without its Transformer (in total or partial removal) is SwinUNet and in the context of architecture, the reason is obvious – it is solely composed of Transformers and has no convolutions in the up and downsampling layers (Table 1). Thus, the segmentation performance of the original SwinUNet, while rather low, is solely due to its Transformer blocks. This highlights, that Transformers are also *capable* of learning usable representations for medical image segmentation without any convolutions.

# 4 BEYOND PERFORMANCE: DO TRANSFORMERS ALTER MODEL BEHAVIOR?

While performance is generally the core metric of interest during development of a new deep architecture, it is not the only one. Aside from performance changes, the addition of Transformer blocks can also lead to altered model behavior. To investigate this, we compare pairs of original and modified networks, with and without the Transformer block, in terms of: **a)** Similarity of segmentation errors between predictions using volumetric error overlap, and **b)** Similarity of learnt representations.

## 4.1 SIMILARITY OF SEGMENTATION ERRORS

Comparing 2 models using DSC with a given input can quantitatively lead to similar values, on very different predictions, since it disregards the specific positions in the input spaces of 2 models where prediction errors are made (Reinke et al. (2023)). To measure similarity between the predictive behavior of networks more closely, we propose the **Volumetric Error Overlap (VEO)**: Let $\hat{Y}_1$ and $\hat{Y}_2$ be the map of predictions of two models $m_1$ and $m_2$ for a single sample and groundtruth $(\boldsymbol{X}, \boldsymbol{Y}) \in \mathbb{D}$ ,with $\mathbb{D}$ being the dataset. We calculate the binary error masks $\boldsymbol{E}_1$, $\boldsymbol{E}_2$ through:

$$\boldsymbol{E}_{w,h,d}(\boldsymbol{Y}, \hat{\boldsymbol{Y}}) = \begin{cases} 1, & \text{if } \boldsymbol{Y}_{w,h,d} \neq \hat{\boldsymbol{Y}}_{w,h,d} \\ 0, & \text{otherwise.} \end{cases} \tag{1}$$

We calculate the pairwise DSC between these error maps, as **VEO** between models

$$\rho(m_1, m_2)_{\mathbb{D}} = \frac{1}{|\mathbb{D}|} \sum_{x \in \mathbb{D}} 2 \frac{|\boldsymbol{E}_1 \cap \boldsymbol{E}_2|}{|\boldsymbol{E}_1| + |\boldsymbol{E}_2|}, \tag{2}$$

which is constrained to values between $\rho(m_1, m_2)_{\mathbb{D}} \in [0, 1]$, with 0 representing 2 models with disjoint error maps and 1 when perfectly overlapping. The VEO measure (Table 2) on its own is difficult to interpret, as it can be confounded by large changes in model performance. Hence, we present it in conjunction with the model performance subject to whole Transformer block replacement. We categorize the role of a Transformer in these architectures based jointly on the similarity of accuracy and errors in 4 categories:

- **Underutilized Transformer (VEO** > 0.95, **P$_{sim}$** > 0.95**):** TransUNet and TransBTS, which both have Transformers in the bottleneck of a UNet, show little difference without the Transformer, indicating an ineffecient architecture design. This is further explored in §5.3.

- **Compensable Transformer** (0.85 ≤ **VEO** < 0.95, **P$_{sim}$** > 0.95**):** Architectures such as CoTr, TransFuse, SwinUNETR and UTNet lose little accuracy without the Transformer but demonstrate a moderate difference in VEO. This indicates that in the absence of the Transformer, their ConvNet backbone can learn meaningful representations to compensate.

- **Non-compensable Transformer** (0.7 < **VEO** < 0.85, **P$_{sim}$** > 0.90**):** In UNETR, the ViT encoder representations are forced through the remaining backbone - when the ViT is absent, identity replacements lead to less powerful features for the remaining backbone. As such the *Transformer helps in learning* non-compensable features but at the cost of performance as a ViT encoder limits feature resolution to the remaining network. nnFormer, on the other hand, uses an alternating Swin and ConvNet design and significantly alters representation learning when absent (§4.2), thus leading to differences in errors.

- **Critical Transformer (VEO** < 0.7, **P$_{sim}$** < 0.90**):** SwinUNet is a pure Transformer design which collapses without its Swin blocks, a criticality which renders this comparison obsolete.

## 4.2 REPRESENTATIONAL CHANGE

While VEO allows us to measure how the output changes, it does not allow insights into the representations of the architecture leading to this output. Hence, we compare the change of internal representations for all 9 architectures when removing the whole Transformer. We compare the representations of layers between models using centered kernel alignment (CKA) (Kornblith et al., 2019). More precisely, we use minibatch CKA from Nguyen et al. (2020) which utilizes the unbiased HSIC of Song et al. (2012) as in Eqs. (3) and (4).

$$\text{CKA}_{minibatch}(\mathbf{K}, \mathbf{L}) = \frac{\frac{1}{k} \sum_{i=1}^{k} HSIC(K_i, L_i)}{\sqrt{\frac{1}{k} \sum_{i=1}^{k} HSIC(K_i, K_i)} \sqrt{\frac{1}{k} \sum_{i=1}^{k} HSIC(L_i, L_i)}} \tag{3}$$

$$\text{HSIC}(\mathbf{K}, \mathbf{L}) = \frac{1}{n(n-3)} \left( tr(\tilde{\mathbf{K}}\tilde{\mathbf{L}}) + \frac{\mathbf{1}^T \tilde{\mathbf{K}} \mathbf{1} \mathbf{1}^T \tilde{\mathbf{L}} \mathbf{1}}{(n-1)(n-2)} - \frac{2}{n-2} \mathbf{1}^T \tilde{\mathbf{K}} \tilde{\mathbf{L}} \mathbf{1} \right) \tag{4}$$

with $\mathbf{L}_i = \mathbf{X}_i \mathbf{X}_i^T$ and $\mathbf{K}_i = \mathbf{Y}_i \mathbf{Y}_i^T$ being composed of the activations of a mini-batch $\mathbf{X}_i \in \mathcal{R}^{n \times p_x}$ and $\mathbf{Y}_i \in \mathcal{R}^{n \times p_y}$. In our experiments, $p_{x/y}$ is shaped either spatially with channel, width, height and

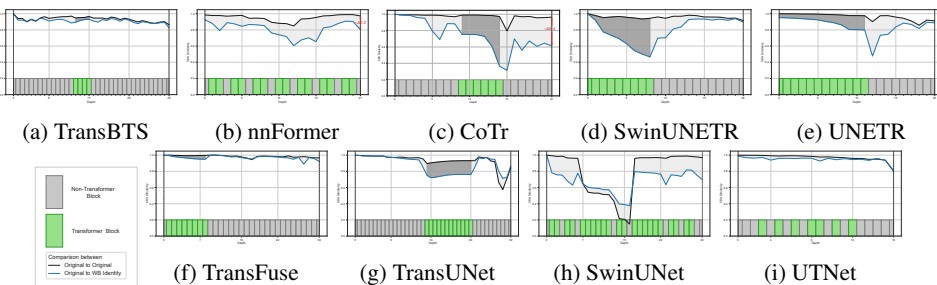

Figure 2: **Some Transformers affect learned representations little more than an identity mapping, while others have a stronger impact.** We visualize the CKA similarity gap (gray) between baseline similarity (black) and similarity between original and networks with replacement (blue). Additionally, we highlight which layers are Transformer blocks (green).

depth dimensions or has a sequence shape of heads, tokens, depth, which is flattened for comparison. We chose to focus on the diagonal of the CKA matrix, as it allows for clear visualization and interpretation. Additionally, we measure the similarity along the 'outer parts' of the UNet design that all architectures follow, ignoring all layers that can affect a skip connection, allowing us to represent these non-sequential models more easily (see Fig. 7). A more detailed explanation of design decisions and hyperparameters of the experiment is provided in §D.

We use this framework to compare the 3 different folds of the original architectures to each other to establish a baseline of representational similarity and subsequently measure the representational similarity between the original architecture and the architecture with whole Transformer blocks replacement (see Fig. 2). In these visualizations, we examine two specific phenomena:

- **Increasing similarity gap during Transformer layers:** Architectures that utilize their Transformer blocks well, change the representations in a non-linear way. When replaced with an identity map, representations do not change in these layers, so the gap (area marked in gray in Fig. 2) between the representational similarity should widen from the beginning to the end of *productive* Transformer layers. If the change of the similarity gap is approximately 0, it indicates that hardly any representational change occurs during the Transformer blocks.

- **Similarity gap at the output of the network ($\triangle$):** Given that the similarity gap widens during the Transformer blocks, we would like the remaining *ConvNet backbone* to not close the similarity gap. If the similarity gap closes, it indicates that the remaining ConvNet can learn very similar representations by itself, indicating a compensable Transformer block.

When inspecting the results visualized in Fig. 2, we can classify our architectures into four categories: A) **Constant gap during Transformers:** TransBTS and TransUNet both exhibit an approximately constant similarity gap, indicating that the Transformer blocks are underutilized and do not change representations. This aligns with the results of §4.1 where we also observed hardly any change in output behavior or predictive performance. B) **Increasing gap during Transfomer but no/small gap at output:** SwinUNETR and UNETR express a strong widening in the similarity gap during their Transformer blocks, indicating that the blocks alter representations, while TransFuse does so only slightly, indicating mild changes. C) **Increasing and maintained gap:** CoTr exhibits a strong widening of the similarity gap that is maintained till the end, indicating its Transformer changes representations in a way that convolutions can not compensate for. nnFormer is harder to interpret due to its alternating block structure, but features an similarity gap at the output, hence we believe both Transformers alter representations strongly in an incompensable way. D) **Did not qualify:** We decided to not classify SwinUNet or UTNet to a prior category, yet show their results for completeness. SwinUNet loses too much performance when removing the Transformer, complicating interpretation. UTNet's similarity gap cannot be determined due to the alternating structure and probably low Transformer influence.

## 5 NATURAL VS. MEDICAL IMAGES: QUANTIFYING THE DOMAIN GAP

Most architectures introduced to the medical imaging domain are inspired by the field of natural images. While adapting advancements from related fields is reasonable, these methods may not translate well due to inter-domain differences. Consequently, we try to quantify the domain gap for two key properties Transformers are known for: A) High data demand and B) Enabling long-range interactions. The first one being an open secret, with the latter one possibly representing an additional roadblock for the success of Transformers.

### 5.1 THE DATA DOMAIN 'CHASM'

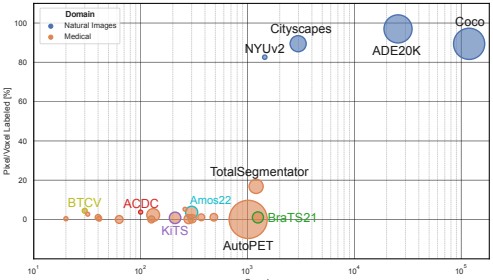

| | BraTS | BTCV | ACDC | # Others |
|---|:---:|:---:|:---:|:---:|
| TransBTS | ✓ | ✗ | ✗ | ✗ |
| TransFuse | ✗ | ✗ | ✗ | 4 |
| TransUNet | ✗ | ✓ | ✓ | ✗ |
| UNETR | ✓ | ✓ | ✗ | 1 |
| UTNet | ✗ | ✗ | ✗ | 1 |
| CoTr | ✗ | ✓ | ✗ | ✗ |
| nnFormer | ✓ | ✓ | ✓ | ✗ |
| SwinUNet | ✗ | ✓ | ✓ | ✗ |
| SwinUNETR | ✓ | ✗ | ✗ | ✗ |
| # Samples | 1251 | 30 | 100 | - |
| Type of Data | Brain Tumor | Organs | Heart | - |

Figure 3: **Medical image segmentation datasets are significantly smaller and sparsely-labeled compared to their natural image counterparts.** Our dataset visualization (Left) illustrates this chasm by the *Average Percentage of Image/Volume Labeled* vs. *Number of Samples* of datasets from both domains. Radii visualizes pixel/voxels over the whole dataset. However, the original evaluation of our 9 Transformer-based models (Right) shows repeated usage of these same small datasets.

Transformer architectures are difficult to train from scratch on small scale datasets, regardless of the domain (Liu et al., 2021a). Therefore pre-training on large datasets is preferred for large Transformer networks even in the natural image domain (Dosovitskiy et al., 2020). The datasets commonly used for this are ImageNet1k with 1.3M images (Russakovsky et al., 2015), ImageNet21k with 14M images (Sun et al., 2017) or even larger proprietary datasets like JFT-300M with 303M images. The realm of medical image segmentation stands in stark contrast to this. Due to the lack of prominent, monolithic architectures and huge datasets that work well for the heterogeneous downstream tasks, almost all models are trained from scratch. The datasets are commonly of small scale, featuring only 10s or 100s of samples (Litjens et al. (2017), Li et al. (2021)). Complicating it further, the samples tend to be sparsely annotated, containing only annotations for a few classes of interest – while natural imaging segmentation datasets tend to be largely fully-labeled.

More recently the TotalSegmentator dataset (Wasserthal et al., 2022) and multi-dataset training (Ulrich et al., 2023) have taken a step in the right direction, tackling the data-sparsity that plagues the medical image domain. We demonstrate this severe chasm between datasets of the medical and natural image segmentation domain in Fig. 3 (Left) by contrasting them by their *number of samples* and their *average fraction of annotated foreground* in each sample. The low dataset size and annotation-sparsity pose substantial difficulties when training architectures in the medical domain. While some Transformer backbones of TransFuse, SwinUNet and TransUNet are pre-trained on ImageNet, the majority of performant architectures – UNETR, CoTr, SwinUNETR, nnFormer, UTNet and TransBTS – train from scratch, with some being trained on BTCV, a dataset comprised of 30 samples. This highlights that data size restrictions native to the medical image segmentation domain are a roadblock to outperforming CNNs with Transformer-based architectures.

### 5.2 PURE CONVNETS ARE MORE 'DATA-EFFICIENT'

Given the large benefits of large scale pre-training of Transformers in the natural imaging domain and the limited nature of medical data, we aim to analyze the influence of data scarcity on Transformer-based networks. We do this by artificially shrinking the size of the training dataset for both organ (AMOS) and pathology (KiTS19) segmentation tasks and measuring segmentation performance of

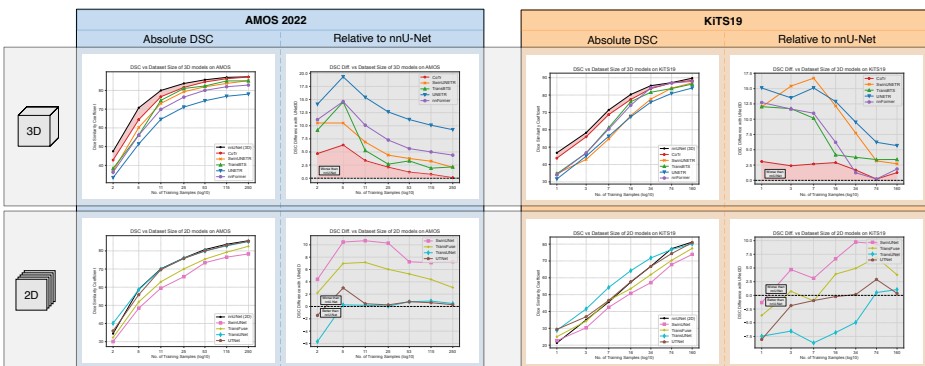

Figure 4: **In a low data regime, 3D CNNs are seen to be better, but with growing data this gap closes showing promise for even larger datasets.** Measuring the DSC of all networks over a large range of training dataset sizes demonstrates that Transformers (particularly 3D) narrow their performance gap to the 3D CNN baseline. For the 2D transformer architectures, performance remains lower than 3D, and they do not improve upon their 2D CNN counterpart in higher data regimes.

our architectures. We train all models with samples ranging from 1%-100% of the total number of training data and evaluate on a held-out test set. (More details in §B.2).

In Fig. 4, we observe the general *but expected* trend that all architectures improve in segmentation performance with increased training data. However, plotting the difference between the UNet baseline (`nnUNet`) and the Transformers highlights a performance gap. For all five 3D architectures, a wide performance gap is seen in the 5-25 samples region, with Transformer-based architectures adapting poorly to reducing the training set size. This gap indeed reduces when the training set size gradually increases to 100%. This is less pronounced in 2D networks with 3 out of 4 architectures showing this phenomenon on AMOS but not on KiTS19.

The performance dynamics highlight an important fact which, although implicitly understood, is often ignored in the medical domain - *Transformers require significantly more data than convolutions*. Therefore, hybrid architectures are likely to be worse than pure ConvNets in low data settings. In light of this, researchers should take dataset size into consideration when choosing architectures – with ConvNets being preferable in low-data scenarios and Transformers for higher data regimes.

In the context of natural images, the relevance of *such* small datasets as in Fig. 4 maybe slightly unrealistic. But as shown in §5.1 - for medical datasets, *it can be the norm*. The Synapse Organ Segmentation dataset (BTCV) (Landman et al. (2015)), overwhelmingly used in 5 out of 9 architectures under evaluation in this work (Fig. 3) contains 30 training samples (and even less after a training-validation split). An example of **'large'** sample size for medical images, the ACDC (Automated Cardiac Diagnosis Challenge) dataset (Bernard et al., 2018) which was also used originally to evaluate a number of networks in this work, contains 100 training samples. This contextualizes the relevance of our conclusions regarding the effectiveness of standard Transformer-Nets in the face of reduced (*or rather, 'expected'*) training set sizes in medical image segmentation.

## 5.3 THE NEED FOR LONG-RANGE INTERACTIONS

Aside from needing a lot of data, Transformers are known to enable long-range interactions. Given this strength, we explore the importance of this attribute in dense 3D medical segmentation. We probe this by slowly reducing the receptive field, for a convolutional medical segmentation network by cutting away downsampled stages of a U-Net. For this we use a 3D nnU-Net on AMOS, as highlighted in Table 3. We observe that in the medical domain, the task is still close to its best performance with a constrained receptive field of [32, 68, 68], and loses 10 DSC points when having a receptive field of [14, 32, 32] voxels, which is as large as CIFAR10/100. This result is especially informative given that all 9 Transformer-based architectures emphasize their proficiency in modeling long-range dependencies. While Transformers-in-bottleneck architectures like TransUNet and TransBTS might particularly struggle, we note that all 9 networks are possibly susceptible to diminishing benefits of capturing long-range interactions in medical image segmentation datasets.

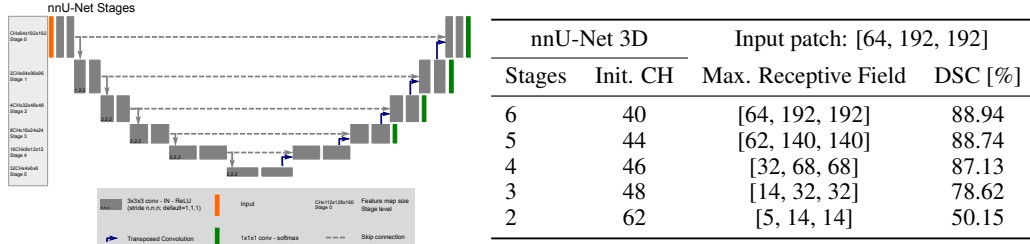

| nnU-Net 3D | | Input patch: [64, 192, 192] | |
|---|---|---|---|
| Stages | Init. CH | Max. Receptive Field | DSC [%] |
| 6 | 40 | [64, 192, 192] | 88.94 |
| 5 | 44 | [62, 140, 140] | 88.74 |
| 4 | 46 | [32, 68, 68] | 87.13 |
| 3 | 48 | [14, 32, 32] | 78.62 |
| 2 | 62 | [5, 14, 14] | 50.15 |

Table 3: **Semantic segmentation can be learned well, despite a constrained receptive field.** We reduce the receptive field of a medical image segmentation architecture, to probe the importance of long-range interactions. Even with a limited receptive field spanning [32, 68, 68] voxels the medical segmentation network task can be learned, indicating low relevance of long-range interactions.

While this experiment yields an intriguing finding, it is essential to acknowledge its inherent limitations. We have focused exclusively on a single dataset and a single UNet architecture. Consequently, our conclusions are specific to the task of organ segmentation within the AMOS dataset, suggesting that, on average, long-range interactions may not be essential for the organs in this particular dataset. Furthermore, it is important to note that in this experiment we reduce the receptive field through the removal of stages, which also leads to a decrease in the overall depth of our architecture. Consequently, the reported values can be interpreted as a lower bound for relevance rather than an explicit measure of it, due the confounder of depth. Despite this, we believe that it is interesting to highlight owing to the emphasis often placed on learning long-range dependencies in medical image datasets.

# 6 DISCUSSION, LIMITATIONS AND CONCLUSION

In this study, we conducted an extensive analysis of existing Transformer architectures in the context of 3D medical image segmentation. Our objective was to identify the most promising Transformer mechanisms for this task. We observed a common trend across most architectures, where a substantial *ConvNet backbone* was often present and performed competitively even without any Transformer blocks. Further investigation revealed that certain architectures, such as nnFormer and CoTr, incorporated Transformers to learn distinct representations but did not yield significant performance improvements over their fully convolutional counterparts. Moreover, our exploration into the domain gap highlighted that several factors may contribute to this phenomenon. Firstly, the stark contrast in dataset sizes between medical and natural image segmentation datasets, coupled with the challenge of training from scratch, could partially explain the limited performance gains from Transformers. Secondly, it is possible that long-range information integration, a strength of Transformers, may have limited relevance in the medical domain.

We strongly advocate for addressing these limitations through two potential avenues. One is the expansion of dataset sizes, either by amalgamating existing datasets (Ulrich et al., 2023) or by employing semi-automatic labeling techniques (Wasserthal et al., 2022). The other avenue is the development of effective self-supervised pre-training schemes (Tang et al., 2022), which we believe could be a pivotal step in unlocking the full potential of Transformer architectures for medical image segmentation. It is worth noting that training Transformers is known to be sensitive to hyperparameters, more so than CNNs. While additional optimization of the Transformer networks could potentially yield marginal improvements in absolute performance, we believe that this limitation does not significantly impact the core findings and takeaways of our study.

In conclusion, this study provides a comprehensive overview of the current state of Transformer architectures in the relatively specialized domain of medical image segmentation. Through our work, we aim to shed light on the existing roadblocks and offer actionable pathways for the development of improved algorithms. We hope this work paves the way towards Transformer architectures that not only compete with but also surpass the current gold standard represented by CNNs in the realm of 3D medical image segmentation.

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
