

Figure 5: MICCAI challenges categorized by their task. Since a long time at least 50% of challenges only focus on semantic segmentation with other tasks being significantly less represented.

## A  The Significance of Semantic Segmentation in 3D Medical Image Analysis

Semantic segmentation plays a pivotal role in the field of 3D medical image segmentation, offering crucial insights and enabling precision in diagnosis and treatment planning. In this section, we explore the profound importance of semantic segmentation, particularly in the context of MRI and CT data, and its far-reaching implications for healthcare.

**Native Data Representation for MRIs and CTs:**  Semantic segmentation serves as the native data representation for 3D medical images obtained through MRI and CT scans. These imaging modalities provide detailed, volumetric views of internal anatomical structures, tissues, and pathologies. Semantic segmentation effectively assigns labels to each pixel or voxel, translating raw data into clinically meaningful information. This conversion is vital in bridging the semantic gap between volumetric data and actionable insights, making it easier to integrate computer vision techniques into medical workflows.

**Crucial Role in Diagnosis and Treatment Planning:**  The accuracy of diagnosis and the effectiveness of treatment planning are of paramount importance in healthcare. Semantic segmentation plays a pivotal role in achieving these goals. By precisely delineating anatomical regions and pathological anomalies, it provides clinicians with a comprehensive understanding of a patient's condition. For example, in oncology, semantic segmentation assists in the localization and quantification of tumors, supporting staging and treatment evaluation. In neurology, it aids in identifying brain structures for precise surgical planning, while in cardiology, it helps assess cardiac chambers and vessels, contributing to cardiovascular health evaluation.

**Advancements in Personalized Medicine:**  Furthermore, the integration of semantic segmentation in 3D medical image analysis aligns with the shift towards personalized medicine. It enables the extraction of patient-specific anatomical and pathological information, facilitating the customization of treatment plans. Additionally, it allows for longitudinal studies by tracking disease progression over time. By harnessing semantic segmentation, healthcare practitioners can offer tailored interventions that optimize patient outcomes, while minimizing risks and side effects. This personalized medicine approach is poised to transform healthcare, providing more effective treatments tailored to individual patients.

**Semantic Segmentation Challenges at MICCAI:**  A significant testament to the importance of semantic segmentation in the medical imaging community is reflected in the annual MICCAI (Medical Image Computing and Computer Assisted Intervention) conference. A vast majority of challenges and competitions at MICCAI revolve around semantic segmentation (see Fig. 5. Imaginary Figure 1 illustrates the dominance of semantic segmentation challenges at the MICCAI conference, highlighting the central role it occupies in advancing the field of medical image analysis.

In summary, semantic segmentation serves as a cornerstone in 3D medical image analysis, particularly in the context of MRI and CT data. Its native representation, support in diagnosis and treatment planning, and contributions to personalized medicine are instrumental in reshaping healthcare. The synergy between computer vision and medical imaging, driven by semantic segmentation, holds promise for improving patient care and catalyzing transformative advancements in 3D medical image segmentation.

## B  TRAINING DETAILS

### B.1  ARCHITECTURE TRAINING HYPERPARAMETERS

The network training scheme is based heavily on the default settings of the nnUNet framework with minor changes added on top of the nnUNet framework (Isensee et al. (2021)). To maintain comparability, the patch size was set to $96 \times 96 \times 96$ for all 3D networks and $512 \times 512$ for all 2D networks (except SwinUNet whose adherence to the Swin Transformer architecture restricted us to $224 \times 224$). The AdamW optimizer (Loshchilov & Hutter (2017)) was used as the optimizer with $1e - 4$ as the learning rate for all ViT-based networks and $5e - 4$ as that of all Swin-based networks. An exception is SwinUNet which showed unstable training performance with $5e - 4$ and thus needed a lower learning rate of $1e - 4$. Table 4 provides a detailed description of the training settings.

| Epochs | Learning Rate | Weight Decay | Optimizer | Data Augmentation | Patch Size | Used nnUNet |
|---|---|---|---|---|---|---|
| SwinUNet[2D] | | | | | $224 \times 224$ | $\times$ |
| TransFuse[2D] | | | | | | $\times$ |
| TransUNet[2D] | $1e - 4$ | | | | $512 \times 512$ | $\times$ |
| UTNet[2D] | | | | | | $\times$ |
| CoTr[3D] | | | AdamW | nnUNet Default[†] | | $\checkmark$ |
| SwinUNETR[3D] | $5e - 4$ | $3e - 5^\dagger$ | | | | $\times$ |
| TransBTS[3D] | $1e - 4$ | | | | $96 \times 96 \times 96$ | $\times$ |
| UNETR[3D] | | | | | | $\times$ |
| nnFormer[3D] | $5e - 4$ | | | | | $\checkmark$ |
| nnUNet[3D] | $1e - 2^\dagger$ | | $SGD^\dagger$ | | | $-$ |
| nnUNet[2D] | | | | | $512 \times 512$ | $-$ |

Table 4: The training details of all networks are provided. The hyperparameters are constant for training during all experimental modes - low dataset experiments or network modification experiments. Some hyperparameters are the default settings[†] of the nnUNet framework.

We maintain consistency in hyperparameters whether we are using these architectures in network modification for isolating the ConvNet backbone (§3) or our low dataset experiments (§5).

### B.2  LOW DATASET EXPERIMENTS

In our experimental design, we maintain a consistent set of hyperparameters, which we use across all experiments (see '§B.1'). As we want to explore the impact of varying dataset sizes on the performance of machine learning models we artificially augment dataset size. We achieve this by systematically creating subsets of the complete training samples drawn from the AMOS and KiTS datasets. For the AMOS dataset, we work with subsets consisting of 2, 5, 11, 25, 54, 116, and 250 samples, corresponding to 1%, 2.1%, 4.6%, 10%, 21.5%, 46.4%, and 100% of the original 250 samples, respectively. For the KiTS dataset, our subsets consist of 1, 3, 7, 16, 34, 74, and 160 samples of the original 160 samples, respectively.

In our approach, we systematically reduce the size of each training subset. We achieve this reduction by randomly discarding 53.6% of the samples from the larger training subset to create the subsequent smaller subset. Importantly, we ensure that each larger subset encompasses all the samples from the preceding smaller subsets, maintaining data continuity throughout this gradual downsizing.

To improve the reliability of our experiments, we use a three-fold cross-validation strategy for each data percentage. These cross-validation folds are consistently applied across all our architectural experiments. We do this to make it easier to compare the results between experiments and to reduce the impact of random sample selection on our findings. This becomes especially important when we have limited data, where the quality of training samples can vary significantly.

Overall, our experimental design offers a robust and systematic means of assessing model performance across a spectrum of data availability conditions, ranging from scenarios with severely constrained datasets to scenarios utilizing the complete dataset. This comprehensive analysis sheds light on how machine learning models perform under varying data constraints.

### B.3 LONG-RANGE DEPENDENCIES EXPERIMENT

In our long-range dependency experiment, we employ nnU-Net v2 and adhere to the native nnU-Net training methodology, which involves 5-fold cross-validation on the AMOS dataset consisting of the full 300 training samples, partitioned using an 80-20 split Isensee et al. (2021). Aside from this we preserve the conventional nnU-Net training procedure as outlined in the seminal work by Isensee et al. Isensee et al. (2021).

To conduct our receptive field experiment, we take a stepwise approach by removing entire stages from our architecture, as illustrated in Figure Table 3 in the main manuscript. We begin with all stages intact and progressively eliminate the lower stages, starting from stage 5 and ending with just stages 0 and 1 remaining in the architecture. This process not only affects the receptive field but also significantly reduces the overall depth of the model, resulting in a substantial decrease in the total number of non-linear operations applied to the data.

It is important to note that, at this point, we choose not to compensate for the reduction in depth by adding 1x1x1 convolutions. This decision is due to the complexity involved in determining the optimal locations for such convolutions, which could introduce unintended complications into our experiments.

Given these practical considerations, we opt to stick as closely as possible to the original nnU-Net architecture while primarily adjusting the number of channels. This approach helps us maintain a consistent VRAM profile of around 10.7GB, ensuring that our experiments can be conducted on a single RTX2080TI.

## C TRANSFORMER NETS VS BASIC UNETS

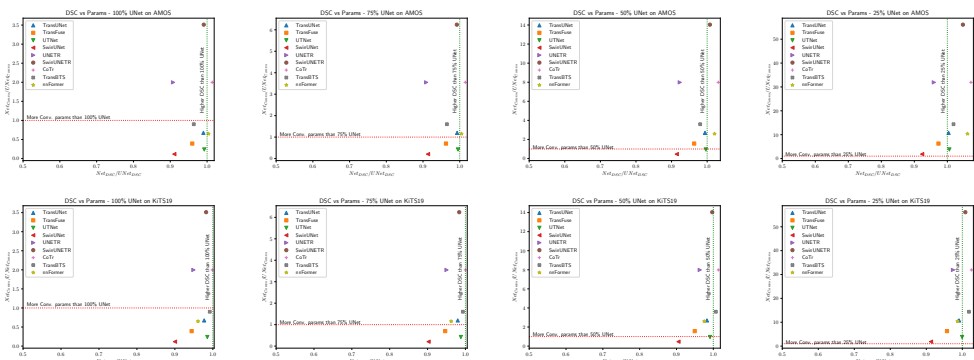

Figure 6: Transformer Nets vs original UNets of widths of 25%, 50%, 75% and 100%

## D REPRESENTATION SIMILARITY EXPERIMENT DETAILS

In this section we explain all the steps conducted for the representational similarity experiments.

**Dataset preparation for representational similarity comparison** Medical image segmentation methods tend to be unable to process the whole 3D volume of a single patient, instead a patch-wise approach is undertaken to predict an entire patient. Additionally, opposed to natural images, the scans usually have a fixed spacing (e.g. 1x1x1 [mm] istrotropic spacing) that practictioners want to maintain. Subsequently we use the validation cases of AMOS to create a patched dataset, which we

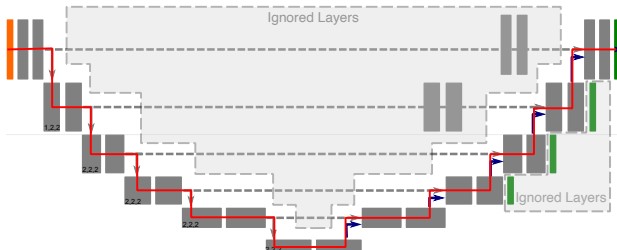

Figure 7: Visualization which positions we select to extract activations from. We select all representations at positions along the red line, after blocks that are not skipped by a residual connection.

use to extract representations on. Since not all architectures share an identical input patch size, we create multiple patched datasets for each input patch size, resulting in one 3D patched dataset with patches of size $96 \times 96 \times 96$ and two 2D datasets of size $224 \times 224$ and one of size $512 \times 512$. We do so by turning off data augmentation on the dataloader used during our training experiments ( see §B), leaving us with a preprocessed region, randomly cropped from the validation case. For each case we extract 5 patches in the 3D case and 25 patches in the 2d case, resulting in patched dataset sizes of 250 for the 3D case and 1250 for the 2D case.

**Representation extraction and comparison**   Given these patches we extract the representations of the architecture along the "outer hull" of the architecture, neglecting potential augmentation internally, to end up with a sequential like structure (see Fig. 7). Additionally we choose to not extract representations when residual connections are present, hence we extract either after a full Transformer block (Fig. 1) or a full Basic residual block or Bottleneck residual block.

**CKA calculation**   Having determined the positions to measure representations and the patched dataset to use for representation extraction, we calculate our mini-batch CKA according to Eq. (3) and Eq. (4). As batch size we chose 64 for all nine architectures. As we have 3 different models for all the experiments we ran, we compare all permutations of the original models to each other, resulting in three similarity values for our baseline similarity (black *'Original to Original'* values). Given the additional 3 models with their Transformer blocks replaced, we compare all 9 combinations of 1 original and 1 replaced model (blue*'Original to WB identity'*).

It may be important to note that all models were trained on the full 250 AMOS training cases, so there was 100% overlap between the training data of all models, with and without replacement.

**Q1: Why do we want a decreasing representational slope?**   We care about whether the Transformer blocks within the architecture do contribute meaningful to the remaining parts of the architecture. Hence we would like the Transformer blocks to change the representations as much as possible from the state they had before the block. When we replace the Transformer block with an identity mapping we guarantee that current representations remain static along the block and no representational change can occur.

Given our representational comparison setting between the original architecture (starring Transformer blocks that can change the representations) and the WB identity architecture (with Transformer blocks that have been replaced with an identity mapping), we want to see that the learned Transformer blocks do something different than an identity mapping.

Should the original architectures Transformer blocks under utilize their Transformer-block no change occurs in them, resembling an identity mapping without constrained to one. This will express itself in the representational similarity staying largely similar for the stretch of the Transformer blocks.

On the other hand, if the architectures utilize their Transformer blocks heavily, it will change the representations a lot, leading to a decrease in similarity to the static baseline with its Transformer blocks replaced by identity mappings.

**Q2: Why is a gap at the output desirable?** When looking at the output similarity we can interpret it as the similarity between the features used for the prediction. Given that this gap is low, we conclude that the learned features are fairly similar, while larger gaps represent less similar features.

Under this light, having replaced the Transformer block with identity mappings and observing no or a small gap, indicates that the final features the architecture without Transformers converged to a similar solution as with Transformers, indicating that the same thing can be learned by convolutions alone. On the other hand observing a large gap indicates that the solutions the architecture with and without Transformers converges to is very different, showing that the features are changed in a way the remaining blocks are not able to achieve by themselves.

We argue that this gap indicates a good use of Transformer blocks, as it adds additional possibilities on how to solve the task, superseding what convolutions can provide by themselves. The low or no gap case instead indicates that the convolutional network can learn the same mapping as the Transformer, so why bother with the high memory demand, more difficult training in a lower data regime, where it is not outperforming convolutions yet?

## E THE RECENT POPULARITY OF TRANSFORMER-BASED ARCHITECTURES IN MEDICAL IMAGE SEGMENTATION

| Network | n-D | Year | Venue | Citations |
|---|---|---|---|---|
| SwinUNet | 2D | 2021 | ECCV(W) | 1181 |
| TransFuse | 2D | 2021 | MICCAI | 499 |
| TransUNet | 2D | 2021 | arXiv | 1969 |
| UTNet | 2D | 2021 | MICCAI | 267 |
| CoTr | 3D | 2021 | MICCAI | 293 |
| nnFormer | 3D | 2022, 2023 | arXiv, IEEE Transactions on Image Processing | 192 |
| SwinUNETR | 3D | 2022 | MICCAI(w) | 261 |
| TransBTS | 3D | 2021 | MICCAI | 366 |
| UNETR | 3D | 2022 | WACV | 726 |
| **Total Citations** | | | | 5754 |

Table 5: **Transformers are popular for medical image segmentation.** Citations over the last 2 years (2021-2023) as of 28.09.2023 show that Transformer-based deep neural networks are increasingly popular for tasks in medical image segmentation.