# OpenReview forum: "Lost in Transformation: Current roadblocks for Transformers in 3D medical image segmentation"
_ICLR.cc/2024/Conference — Submitted to ICLR 2024_

### Official Review · Reviewer_aM1T · 2023-10-30

**Soundness:** 1 poor
**Presentation:** 1 poor
**Contribution:** 1 poor
**Rating:** 1
**Confidence:** 4

**Summary:**

In this work, authors systematically dissect 9 popular hybrid Transformer networks on two representative organ and pathology segmentation datasets and explore whether Transformers are still beneficial under these challenging conditions.

**Strengths:**

The paper is easy to read.

**Weaknesses:**

1.	There is no contribution or novelty. This is more like comparing 9 frameworks.
2.	Experimental design has some flaws. There are so many transformer networks and picking 9 out of those should have a valid assumption, which is missing in the paper.
3.	Lacking literature review.

**Questions:**

1.	Only quantitative analysis is provided in the manuscript. What about qualitative analysis? Showing segmentation masks would help readers to understand which method performs well, especially when it comes to the medical AI domain, qualitative analysis is a must.
2.	A comparison of intermediate attention maps is missing.
3.	Representation similarity can be visualized as a heat map comparing network layers. This would give some understanding to the reader of how feature extraction works and whether it’s similar across all (or part of) networks or not.

---

> ### Author Response · Authors · 2023-11-22
> **Response 1/2**
>
> While we appreciate the difficulties of the reviewing process, we want to note that Reviewer 4 provides little actionable feedback. We also notice strong opinions in stark contrast to other reviewers as well as the accepted opinion of the domain, calling into question a possibly limited familiarity with the medical image analysis domain by the reviewer.
>
> > The paper is easy to read.
>
> We appreciate that the Reviewer found our presentation easily comprehensible. Perhaps, given this strength, one may assume that the ‘presentation’ should probably not be rated as poor.
>
> > There is no contribution or novelty. This is more like comparing 9 frameworks
>
> Novelty is a subjective idea and we believe that we highlight an important downside concerning the domain of medical image segmentation. We took the 9 most popular Transformer-based networks in the domain and demonstrated minimal contribution of the Transformer component. We believe that this should guide research in directions that would better utilize Transformers in the medical image segmentation domain. We respectfully disagree with the notion that this is simply “comparing 9 networks” and would have benefited from the reviewer expanding on this a bit further.
>
> > Experimental design has some flaws.
>
> While the reviewer states that there exist potential flaws in the experiment, it would be more impactful if these concerns were linked to specific sections of the paper. We believe that the inclusion of actionable recommendations for improvement of experimental design would make the feedback more constructive.
>
> >  There are so many transformer networks and picking 9 out of those should have a valid assumption, which is missing in the paper.
>
> Given the strong comment on our supposed random network selection, we urge the Reviewer  to Section 2 (Experimental Design) where we state  that: “We dissect a number of massively-influential Transformer architectures for medical image segmentation, regularly used as blueprints for designing newer architectures or state-of-the-art baselines. In this work, we focus on 9 such networks with 5500+ citations collectively in the last 3 years (see Table 5)” Our motivation in network selection is clearly aimed at the most popular networks in the domain of medical image segmentation which we clearly state.
>
> > Lacking literature review.
>
> Of course, we would have liked to provide a large literature review for the benefit of the Reviewer. However, we are constrained by a 9 page limit in ICLR and urge the reviewer towards the citation of (Xiao et al. (2023); Shamshad et al. (2023)) from our paper which are large reviews on the topic. We ourselves cover the most popular works in the field in our own paper which should be able to provide the Reviewer a detailed idea of Transformer models in medical image segmentation.

---

> ### Author Response · Authors · 2023-11-22
> **Response 2/2**
>
> > Only quantitative analysis is provided in the manuscript. What about qualitative analysis?  Showing segmentation masks would help readers to understand which method performs well, especially when it comes to the medical AI domain, qualitative analysis is a must.
>
> We unfortunately disagree with the notion of qualitative analysis being superior or even necessary. As informed researchers in the medical image analysis domain, we value quantitative analysis highly, whereas qualitative analysis is usually nice-to-have but not mandatory. This is because it is prone to cherry-picking and does not allow for an intuitive measure of segmentation quality, especially not in the 3D domain where predictions are very hard to visualize. While relevant in a more clinical setting, in the medical image segmentation, we severely doubt the veracity of “qualitative analysis being a must-have” in comparing segmentation techniques.
>
> > A comparison of intermediate attention maps is missing.
>
> This is a process standard to Transformers but we believe not relevant to our scenario. If the removal of an entire Transformer shows no performance degradation, we are unsure as to how visualizing intermediate attention maps help. We believe that our Transformer ablation is a much stricter demonstration of Transformer Utilization by our networks that the one requested by the reviewer.
>
> > Representation similarity can be visualized as a heat map comparing network layers. This would give some understanding to the reader of how feature extraction works and whether it’s similar across all (or part of) networks or not..
>
> Tying to the previous point, we value quantitative results over qualitative results. With that in mind, we deliberately chose to not use a heatmap visualization of the similarity but the diagonal values as line-plots. This allows readers to read explicit, quantitative values and differences from the plot, whereas in a heatmap the reader would have to interpret changes due to draw conclusions making for bad interpretability and low added value. We urge the Reviewer to consider that there is little loss of information compared to their preferred style of visualizing representation similarity.

---

### Official Review · Reviewer_Z59e · 2023-11-01

**Soundness:** 2 fair
**Presentation:** 2 fair
**Contribution:** 2 fair
**Rating:** 5
**Confidence:** 4

**Summary:**

Summary: This paper investigates thoroughly recent transformer architecture for medical image segmentation and challenges the recent trend of developing novel transformer architecture. The authors have systematically ablated different key components and studied their effect on the performance. They have concluded that transformers and their long-range dependency modeling are often not the critical components of the architecture.

**Strengths:**

Strengths:

+ Good categorical benchmark on different transformer component

+ Detailed analysis of their performance and representational behavior.

**Weaknesses:**

Major comments:

- While the medical image segmentation task, the utility of the transformer, can be brought under scrutiny, this is not true for panoptic/instance segmentation and video segmentation, not only because of the data set size but also because of the fundamental difference in network architectures. This makes the criticism of transformers very specific to U-net-like models popular in the medical imaging community, which makes the paper relatively less appealing to the general image segmentation community.

- While the paper quite convincingly points out flaws in the current practice of architectural design in medical image segmentation, the paper did not bring any new ideas to mitigate the issue, which remains a  major weakness and is hard to address within the rebuttal period. Hence, despite being a good review and investigative paper, I am not sure whether it is a good fit for ICLR.

- The use of volumetric error overlap is confusing in concluding model behavior. Given that all models considered provide points estimate, how can the author assert that the apparent difference in model behavior is not a result of the underlying uncertainty? It will be good to know the volumetric error map between three runs of the same models as a reference to the uncertainty because the authors took the same approach for representational change measurement.  And how are the thresholds 0.95, 0.85, etc. chosen? Seems quite arbitrary.

**Questions:**

see weaknesses

---

> ### Author Response · Authors · 2023-11-22
> **Response to Reviewer Z59e**
>
> We thank the reviewer for their valuable feedback and attempt to address some of the issues raised below.
>
> > While the medical image segmentation task, the utility of the transformer, can be brought under scrutiny, this is not true for panoptic/instance segmentation and video segmentation, not only because of the data set size but also because of the fundamental difference in network architectures. This makes the criticism of transformers very specific to U-net-like models popular in the medical imaging community, which makes the paper relatively less appealing to the general image segmentation community.
>
> We agree with the reviewer that our findings do not generalize to other domains like e.g. video segmentation, where data is less hard to come by, but we should be absolutely clear that we also do not claim so and constrain ourselves to the domain of 3D medical image segmentation. We believe that 3D medical image segmentation, while very specific as a domain, is also extremely large and that our findings will significantly influence a plurality of of researchers in this area. We hope that the reviewer will consider this and not use this as a determining factor that influences the score of the paper.
>
> > While the paper quite convincingly points out flaws in the current practice of architectural design in medical image segmentation, the paper did not bring any new ideas to mitigate the issue, which remains a major weakness and is hard to address within the rebuttal period. Hence, despite being a good review and investigative paper, I am not sure whether it is a good fit for ICLR.
>
> The reviewer is correct that we do not provide any new ideas to mitigate the issue, yet we hope to highlight the issues in order to shift the focus from currently _proposing new architectures_ to _proposing effective architectures_ - we hope our research demonstrating the weaknesses of Transformers will enable researchers to improve them further. We also address this further in the general comments for the rebuttal which you may find at the top of this page.
>
> > The use of volumetric error overlap is confusing in concluding model behavior. Given that all models considered provide points estimate, how can the author assert that the apparent difference in model behavior is not a result of the underlying uncertainty? It will be good to know the volumetric error map between three runs of the same models as a reference to the uncertainty because the authors took the same approach for representational change measurement. And how are the thresholds 0.95, 0.85, etc. chosen? Seems quite arbitrary.
>
> The reviewer echoes our sentiments exactly about the underlying uncertainty. The VEO value is indeed a ratio of:
> 1) 3 runs of the same model as reference
> 2) all permutations of these models with the Transformer removed.
>
> The same approach as representation change measurement was taken here as well. We can make this explicit in the text and apologize for not doing so. The thresholds themselves are formulated by jointly considering VEO and P_sim and their joint degradation across the plots in Table 2. They are selected specifically at points of degradation of one or both scores on the plot and are for illustrative purposes.

---

### Official Review · Reviewer_BnaE · 2023-11-05

**Soundness:** 2 fair
**Presentation:** 2 fair
**Contribution:** 2 fair
**Rating:** 5
**Confidence:** 3

**Summary:**

The study analyzed the effectiveness of nine Transformer-based models for segmenting medical images, focusing on two datasets centered on organ and pathology segmentation. It was found that convolutional layers are essential to these models' performance, whereas the transformer layers may not be as vital. Additionally, the researchers questioned the assumed significance of long-range dependencies—a characteristic feature of transformer models—in the context of medical image segmentation.

**Strengths:**

In a probing study on the role of transformer models within medical imaging segmentation, the authors challenged their utility compared to traditional CNN architectures. They conducted a comparative analysis of nine cutting-edge architectures by substituting transformer blocks within these models. Their research unveiled interesting results, revealing minimal performance disparity between the original and modified models. This suggests that the transformer's capability may be underutilized in the medical imaging segmentation tasks evaluated.

**Weaknesses:**

The study's scope was confined to a narrow selection of segmentation datasets, and the ablation studies conducted were restricted in its current form.

**Questions:**

1. The authors' research on medical imaging segmentation with Transformer-based models, focusing on organ and pathology within specific datasets, may not capture the potential benefits of long-range dependencies in all medical imaging scenarios. For example, cardiac video segmentation may reveal different results due to the temporal dynamics involved. The authors could provide insights on whether their findings are applicable to such medical imaging tasks where Transformers might show utility.

2. Can the authors clarify whether the transformer blocks were pretrained with natural image datasets.

3. The study used nnU-Net as a benchmark for evaluating performance. Its status as an industry benchmark relies heavily on advanced data augmentation and meticulous hyperparameter optimization. The nine Transformer-based architectures assessed may not employ these sophisticated techniques. For an equitable comparison, it's crucial that all other variables, such as data augmentation protocols, are standardized across models. Without this uniformity, any observed differences in performance could be attributed to varying methodologies rather than inherent architectural distinctions.

---

> ### Author Response · Authors · 2023-11-22
> **Response to Reviewer BnaE**
>
> We thank the Reviewer for his feedback and want to address the questions raised first.
>
> ## Questions:
>
> **Question 1:**
> > The authors focus on organ segmentation and pathology segmentation specific datasets. They may not capture the potential benefits of long-range dependencies in all medical imaging scenarios. Cardiac video segmentation may reveal different results due to the temporal dynamics involved. The authors could provide insights on whether their findings are applicable to such medical imaging tasks where Transformers might show utility.
>
> **Response 1:**
> In our current experiments we constrain ourselves to 3D organ and pathology segmentation as these are the tasks that these architecures originally were proposed for (Fig. 3 - right side) and as these are the tasks that represent the majority of use-cases, as we show in the appendix A (Fig. 5). We agree that there could be slightly different conclusions if the methods were evaluate on temporal datasets, however, to bring the reviewer’s example to our investigative domain, if for example, one decided to segment objects in cardiac videos ignoring the time axis using these Transformer-based network, our results would hold.
>
> **Question 2:**
> > Can the authors clarify whether the transformer blocks were pretrained with natural image datasets.
>
> **Response 2:**
> In the original papers of the 3D architectures none of the original authors used pre-trained transformers, hence we follow that notion in order to keep our evaluation as close to the original as possible and subsequently do not use pre-trained transformer blocks throughout our experiments with these. Similarly for the 2D architectures different authors follow different protocols with some using and others not using pre-trained ViT transformers. To be precise [SwinUNet](https://arxiv.org/pdf/2105.05537.pdf), [TransFuse](https://arxiv.org/pdf/2102.08005.pdf) and[TransUNet](https://arxiv.org/pdf/2102.04306.pdf) were pretrained on ImageNet with UTNet being trained from scratch. We admit this is a deviation from the official use, yet we deliberately chose this considering a recent article showing that pre-trained weights do not help for segmentation tasks. (See [Rethinking pre-training on medical imaging](https://doi.org/10.1016/j.jvcir.2021.103145))
>
> **Question 3:**
> >The study used nnU-Net as a benchmark for evaluating performance. Its status as an industry benchmark relies heavily on advanced data augmentation and meticulous hyperparameter optimization. For an equitable comparison, it's crucial that all other variables, such as data augmentation protocols, are standardized across models.
>
> **Response 3:**
> It is true that we do leverage nnU-Net as a benchmark for evaluating performance, yet nnU-Net does not rely on extensive hyperparameter tuning. nnU-Net is a framework that determines a single, fixed setting of hyperparameters. It follows a variety of heuristics, as determined by the author, and selects, e.g. an appropriate spacing, patch size and normalization given the statistics of the dataset. As these are dataset specific choices, we use them for all of our experiments. Similarly nnU-Net selects an augmentation schemes, but generally differentiates between two simple, default augmentation schemes, one for 3D data and one for 2D data.
> In our Experiments we do apply the identical spacings, patch sizes, normalization schemes that nnU-Net selects for the 3D U-Net to all architectures.The only difference in the settings is based on learning rate, optimizer and weight decay, where we follow the recommendations of the original papers.
>
> ### Weaknesses
> >The study's scope was confined to a narrow selection of segmentation datasets, and the ablation studies conducted were restricted in its current form.
>
> We agree that only 2 datasets is a limitation of our work, hence we extended our experiments to verify our core results on TotalSegmentator, which is the largest annotated dataset in the landscape of 3D medical image segmentation. However, we want to emphasize that we want to investigate whether transformers are feasible given the data that is present in 3D medical image segmentation. Hence we chose the next two best large, non-trivial datasets representing both pathology and organ segmentation. We chose these large datasets, as they represent the datasets where transformers should bring the largest benefits over CNNs, and as they enable our dataset-size ablations.
>
> | Method      | Standard     | Whole Block Replaced |
> |-------------|--------------|----------------------|
> | CoTr        | 89.88 (0.21) | 89.09 (0.28)         |
> | SwinUNETR   |  87.08 (0.35)  | 85.15 (0.21)         |
> | TransBTS    | 87.74 (0.38) | 86.31 (0.78)         |
> | UNETR       | 80.79 (0.41) | 72.86 (0.56)         |
> | nnFormer    | 85.30 (0.28) | 80.37 (0.14)         |
> | nnUNet (3d) | 89.04 (0.11) |                      |
>
> The missing values are currently being evaluated but we will include them in Table 2 left.

---

### Official Review · Reviewer_kZHs · 2023-11-05

**Soundness:** 3 good
**Presentation:** 3 good
**Contribution:** 2 fair
**Rating:** 3
**Confidence:** 4

**Summary:**

This paper analyzes 9 transformer-based networks for medical image segmentation over two public datasets and shows the limitations of these transformer networks.

**Strengths:**

1. The paper systemically analyzes the roles of transformer encoders in medical image segmentation tasks.

2. The authors show that once more time adding transformers layers blindly is not necessarily linked to superior performance, especially in medical image analysis.

3. The authors conduct quite extensive experiments.

**Weaknesses:**

1. The authors only tested on two public datasets, which might not be convincing enough for an investigative paper to validate the claims.

2. Some of the findings by the authors were already identified in the ViT paper in 2020, such as the Transformer will be better when facing a larger dataset but might be worse when having a small dataset such as in medical imaging.

3. The scope of the paper is more investigative rather than innovative, which makes it look more like a technical report/survey rather than a research paper.

**Questions:**

1. The success of transformers is generally due to having less inductive bias and intuitively any application that does not benefit from such fact might not find having such layers helpful. From the ViT paper, the size of the dataset also matters a lot in showing the performance of transformers. The findings 2) and 3) seem to be the direct translation of the above two points, and thus might not be super novel and meaningful.

2. Observation 2 in section 3.1 might not be too meaningful since convolutional blocks are all removed in up/down sampling paths. Replacing transformer blocks with convolutional blocks might be more fair.

3. What does the tick/cross mean in the right table of Figure 3? It would be more clear to add a description directly in the caption.

4. It would be more interesting to discuss how sensitive the transformers are to hyperparameters and summarize the common practice of selecting a reasonable set of hyperparameters.

---

> ### Author Response · Authors · 2023-11-22
> **Response to Reviewer kZHs**
>
> We thank the Reviewer for his feedback and want to address the questions raised first:
>
> ### Questions
>
> >**Q1: ViT paper already showed the need for a lot of data, so section 2) & 3) are not that meaningful.**
>
> **Response 1:** We agree that the findings in section 2) and 3) are closely related to the ViT paper’s findings that the dataset size matters, however this did neither stop any of the authors to apply transformers to the 3D medical domain, nor did this answer the question ‘Do we have enough data in the 3D medical domain to make transformers work?’ – Since there is a domain gap, 3D volumetric data may be worth ‘more’ hence datasets may be large enough. We show this is currently not the case for any of the prominent architectures on the largest public datasets existing.
>
> >**Q2: Observation 2 in section 3.1 might not be too meaningful since convolutional blocks are all removed in up/down sampling paths.**
>
> **Response 2:** We assume the Reviewer mistyped transformer blocks instead of convolutional blocks, as we do not touch convolutions but only the transformer parts of the architecture. Nonetheless, this seems to be a misunderstanding. In our experiments we do not cut off any of the low-resolution paths with our replacements. For 7 of 9 architectures we did not have to apply any special rules to keep the information flow of up/down sampled branches intact. For the remaining two, nnFormer and UTNet,  we had to add slightly more complex behaviour. For junctions in the upsampling branch of nnFormer we joined the two incoming inputs via addition (in the Windowattention). 2) In UTNet when upsampling happens  we add a 1x1 Conv to project to the correct (lower) channel dimension of the higher resolution featuremaps and follow it by bi/tri-linear upsampling. When designing these manual interventions we tried to keep the complexity of the modules to a minimum while still enabling information flow of all branches coming in.
>
> >**Q3: What does the tick/cross mean in the right table of Figure 3?**
>
> **Response 3:** The ticks in the figure highlight which datasets the authors develop/report/evaluate on in their paper. It highlights that the majority of datasets used in these works are small datasets, which are more prone to noise. We add this to the caption of the figure to convey this more clearly.
>
> >**Q4: It would be more interesting to discuss how sensitive the transformers are to hyperparameters and summarize the common practice of selecting a reasonable set of hyperparameters.**
>
> **Response 4:** We agree that this is also an interesting notion and may contribute to the issue of transformers in the domain. However, we maintain that this is slightly beyond the scope of this work given the diverse nature of the datasets.
>
> ### Regarding the Weaknesses:
>
> >The authors only test on two datasets.
>
> We agree that only 2 datasets is a limitation of our work, hence we extended our experiments to verify our core results on TotalSegmentator, which is the largest annotated dataset in the landscape of 3D medical image segmentation. However, we want to emphasize that we want to investigate whether transformers are feasible given the data that is present in 3D medical image segmentation. Hence we chose the next two best large, non-trivial datasets representing both pathology and organ segmentation. We chose these large datasets, as they represent the datasets where transformers should bring the largest benefits over CNNs, and as they enable our dataset-size ablations.
>
> To show the promised Total segmentator results:
>
> | Method      | Standard     | Whole Block Replaced |
> |-------------|--------------|----------------------|
> | CoTr        | 89.88 (0.21) | 89.09 (0.28)         |
> | SwinUNETR   | 87.08 (0.35)  | 85.15 (0.21)         |
> | TransBTS    | 87.74 (0.38) | 86.31 (0.78)         |
> | UNETR       | 80.79 (0.41) | 72.86 (0.56)         |
> | nnFormer    | 85.30 (0.28) | 80.37 (0.14)         |
> | nnUNet (3d) | 89.04 (0.11) |                      |
>
> The missing values are currently being evaluated but we will include them in Table 2 left.
>
> >Some findings were identified by the ViT paper already.
>
> Please see Response 1
>
> >The paper is more investigative than innovative
>
> Please see our general response.

---

### Author Response · Authors · 2023-11-22
**General Response to all Reviewers**

We thank the Reviewers for providing their opinions on our paper and providing us with opportunities to improve our work. We address their individual concerns in the responses and hope that our added experiments and explanations help clarify any previously unanswered questions. We want to take this shared comment as an opportunity to address the seemingly most
 important point of criticism that Reviewers 1,3 and 4 share and which seems to have had the most collective impact on our given rating. We rephrase it as “The paper highlights problems well, but does not provide a solution to them, hence it is less relevant/does not have a novel contribution”.

Firstly, in the field of ML investigative papers that do not provide a novel method but are investigative are not uncommon and can have high impact. A few high-impact examples of this are:

- [How does batch normalization help optimization? - 1.8k citations](https://proceedings.neurips.cc/paper/2018/hash/905056c1ac1dad141560467e0a99e1cf-Abstract.html);

- [Do Vision Transformers See like Convolutional Neural Networks? - 599 citations](https://proceedings.neurips.cc/paper_files/paper/2021/hash/652cf38361a209088302ba2b8b7f51e0-Abstract.html);

- [Are Convolutional Neural Networks or Transformers more like human vision?](https://arxiv.org/abs/2105.07197)

Secondly, in the 3D medical image segmentation domain, enormous efforts in terms of time and compute go into developing novel methods. These methods are pit against existing baselines and superior performance is demonstrated. Currently, Transformer-based methods seemingly are the techniques to beat. Yet, we clearly demonstrate a flaw in 9 of the most popular baseline methods – they have very minimal Transformer utilization to begin with. This calls into question their utility as baselines, state of the art segmentation techniques as well as skeletons for further architecture development.


However, for practitioners the amount of effort needed to verify the above is not apparent. Many people have and more people will invest a lot of time and waste their limited resources in trying to get the baselines method to work as well as the authors claim it should be, which slows down progress of the field as a whole.

Subsequently we believe that revisiting pre-existing methods and evaluating the contribution of the attention mechanisms in them is an important contribution to the field. It spreads awareness on which architectures are more promising and which less. Further, it highlights that not just the architecture, but also a training scheme that is appropriate for the domain is a must, to enable transformers in the 3d medical image segmentation field.

Having said this we want to draw attention to the official ICLR guidelines:
> What is the significance of the work? Does it contribute new knowledge and sufficient value to the community? Note, this does not necessarily require state-of-the-art results. Submissions bring value to the ICLR community when they convincingly demonstrate new, relevant, impactful knowledge (incl., empirical, theoretical, for practitioners, etc).

Hence we ask the Reviewers to reconsider our work under these aspects. We believe, while not providing a solution to the problem, this paper convincingly provides new, relevant knowledge that is valuable for the practitioners and the field as a whole.

---

### Meta-Review · Area_Chair_Z9bm · 2023-12-11

**Metareview:**

The study questions the assumed importance of Transformers and long-range dependencies in the context of medical image segmentation by systematically examining the effectiveness of nine popular hybrid Transformer networks for medical image segmentation tasks, specifically focusing on organ and pathology segmentation datasets. The authors conduct a comprehensive analysis, dissecting the transformer-based models and evaluating their performance.  The paper's strengths include its readability and the systematic analysis of nine hybrid Transformer networks, providing a comprehensive comparison. The paper is easy to understand, facilitating a clear presentation of the research. However, it's important to note that the identified strengths are limited, and the paper faces significant weaknesses in terms of contribution, experimental design, and novelty according to reviewers' assessments. Reviewers highlight a lack of clear contribution or novelty. The experimental design is criticized particularly in the selection of the nine transformer networks without a valid assumption. The absence of a comprehensive literature review is noted as a weakness, and concerns are raised about the paper's lack of qualitative analysis, intermediate attention map comparisons, and visualization of representation similarity. The pure quantitative comparison can be explicit and clear, but do not show much scientific insight. Overall, the paper is largely criticized regarding the soundness and contribution, leading to a consensus in rejection among all reviewers.

**Justification For Why Not Higher Score:**

While the paper is considered easy to read, there are significant concerns about its contribution, experimental design, and absence of novelty, as evidenced by all 4 reviews.

**Justification For Why Not Lower Score:**

N/A

---

### Decision · Program_Chairs · 2024-01-16

Reject